# Use of removable support boot versus cast for early mobilisation after ankle fracture surgery: cost-effectiveness analysis and qualitative findings of the Ankle Recovery Trial (ART)

Petra Baji [1], Estela C Barbosa [1,2], Vanessa Heaslip [3,4] Bob Sangar,[5] Lee Tbaily,[5] Rachel Martin,[5] Sharon Docherty [6] Helen Allen,[7] Christopher Hayward [8,9] Elsa M R Marques [1,10]

PB and ECB contributed equally.

PB and ECB are joint first authors.

For numbered affiliations see end of article.

**Correspondence to**
Dr Petra Baji;
petra.baji@bristol.ac.uk

## ABSTRACT

**Objectives** To estimate the cost-effectiveness of using a removable boot versus a cast following ankle fracture from the National Health Service and Personal Social Services (NHS+PSS) payer and societal perspectives and explore the impact of both treatments on participants' activities of daily living.

**Design** Cost-effectiveness analyses and qualitative interviews performed alongside a pragmatic multicentre randomised controlled trial.

**Setting** Eight UK NHS secondary care trusts.

**Participants** 243 participants (60.5% female, on average 48.2 years of age (SD 16.4)) with ankle fracture. Qualitative interviews with 16 participants. Interventions removable air boot versus plaster cast 2 weeks after surgery weight bearing as able with group-specific exercises.

**Primary and secondary outcome measures** Quality-adjusted life years (QALYs) estimated from the EQ-5D-5L questionnaire, costs and incremental net monetary benefit statistics measured 12 weeks after surgery, for a society willing-to-pay £20 000 per QALY.

**Results** Care in the boot group cost, on average, £88 (95% CI £22 to £155) per patient more than in the plaster group from the NHS+PSS perspective. When including all societal costs, the boot saved, on average, £676 per patient (95% CI −£337 to £1689). Although there was no evidence of a QALY difference between the groups (−0.0020 (95% CI −0.0067 to 0.0026)), the qualitative findings suggest participants felt the boot enhanced their quality of life. Patients in the boot felt more independent and empowered to take on family responsibilities and social activities.

**Conclusions** While the removable boot is slightly more expensive than plaster cast for the NHS+PSS payer at 12 weeks after surgery, it reduces productivity losses and the need for informal care while empowering patients. Given that differences in QALYs and costs to the NHS are small, the decision to use a boot or plaster following ankle surgery could be left to patients' and clinicians' preferences.

## STRENGTHS AND LIMITATIONS OF THIS STUDY

⇒ We performed a full economic evaluation alongside a definitive multicentre randomised controlled trial in the UK comparing a removable boot with plaster cast for treatment of ankle fractures 2 weeks post-surgery weight bearing as able.

⇒ Our study collected resource use data from 243 trial participants and a review of medical notes at the trial centres to estimate costs from both the National Health Service and Personal Social Services payer and the societal perspectives.

⇒ The study was strengthened by a nested qualitative study to explore the impact of both treatments on participants' activities of daily living.

⇒ The randomised controlled trial (RCT) was not powered to detect a difference in quality-of-life or costs and there is uncertainty in the economic result, with the CI around the incremental net monetary benefit statistics crossing the null.

**Trial registration number** ISRCTN15497399, South Central—Hampshire A Research Ethics Committee (reference 14/SC/1409).

## BACKGROUND

Every year, there are about 20 000 people in England admitted for ankle fractures,[1] affecting people of all ages but majority of working age.[2] This has been associated with long-term disability and pain, resulting in time-off work and other economic consequences.[3 4] According to a recent systematic review, the direct costs of ankle fractures range from $1908 to $19 555 worldwide.[5] In England, the cost of inpatient hospital care for ankle fractures was estimated to exceed £63.1 million in 2016/2017.[1] Costs from the social perspective are even higher due to indirect costs associated with fractures, such as

productivity loss and informal care costs, that contribute to almost half of total costs.[6] A previous meta-analysis suggested that the early return to function should be the main goal following surgical management of fractures,[7] which could also reduce economic impact.

About half of patients with ankle fracture require surgery for ankle stabilisation. These patients are typically immobilised with a plaster cast immediately following surgery. After a couple of weeks, ankle management options are more varied. Another plaster cast can be applied for several more weeks, which allows weight bearing and maximises support, but has been associated with joint stiffness, muscle atrophy and deep vein thrombosis.[8–10] One alternative is to use an air boot that can be removed to enable ankle mobilisation exercises, and for comfort and personal care, but could increase the risk of complications.[10 11] Systematic reviews conclude that well-designed, prospective studies are still needed to determine which immobilisation technique offers the most benefit for treating ankle fractures.[7 12] Furthermore, evidence on cost-effectiveness of boots compared with casts is missing from the literature.[12]

The Ankle Recovery Trial (ART) compared early mobilisation in a removable boot with cast immobilisation following ankle fracture fixation where early weight bearing was encouraged in both groups. Participants were followed from 2 weeks until 12 weeks postsurgery and the trial included a full economic evaluation and nested qualitative study. Clinically, there were no observed differences in ankle function between treatment groups at any time point. Overall, early weight bearing in both the boot and cast was safe with low wound complication rates (7%) that, although more frequent in the boot group, were minor.[13] The aim of this paper is to report the results of the cost-effectiveness and related qualitative analyses to further the evidence base and inform future national guidance on postsurgical ankle management.

## METHODS
### Trial design and ethics approval
A health economic evaluation and qualitative component were included in the ART pragmatic multicentre randomised controlled trial (RCT) to provide a more inclusive picture. The trial was conducted across eight UK NHS secondary care trusts between July 2015 and November 2018 and compared casts with removable support boots, combined with weight bearing as able, as methods of ankle fracture management from 2 weeks postsurgery. Most participants at the 6-week follow-up time point were expected to remove their plaster cast or boot and start to mobilise freely. Questionnaires were completed at randomisation (2 weeks after surgery), then at 6, 7 and 12 weeks postsurgery. The primary clinical endpoint was the Olerud and Molander ankle score (OMAS)[14] measured at 7 weeks postsurgery, when participants should have adapted to mobilising without plaster cast or boot. The ART trial was approved by the South Central—Hampshire A Research Ethics Committee (reference 14/SC/1409) and was prospectively registered (ISRCTN: 15497399). Results on clinical effectiveness (such as primary outcome measure and secondary outcomes, such as OMAS at week 12, EQ-5D visual analogue scale (VAS) scores at weeks 7 and 12, secondary mechanistic measures at 6 weeks, weight bearing status, use of walking aids, return to driving and work, impact on daily activities at weeks 6 and 12, and complications and serious adverse events at 6 and 12 weeks) can be found in the final report to the Funder[13] and will be reported in a separate paper.

### Patient and public involvement
Twelve patient advisors, most of whom had experienced an ankle fracture, advised on several aspects of the trial, including determining the minimum significant difference in the primary outcome, decision to include a qualitative component, refining data collection tools, and on patient facing materials and shaping the interview topic guide. Two advisors also attended the trial steering committee.

### Participants
All adults (>16 years old) undergoing open reduction and internal fixation (ORIF) surgery for an unstable ankle fracture were eligible. Exclusion criteria were open ankle fracture; concern about quality of fixation/wound integrity; requiring further stabilisation in/around the ankle (eg, syndesmosis); active leg ulceration; poor skin condition at operating site; serious concomitant disease, diabetic/other sensory neuropathy; non-ambulatory prior to injury; inability to understand/complete outcome questionnaires; enrolment in other interventional research which may confound data collection and concomitant injuries which may affect rehabilitation.

### Intervention and usual care
Participants were randomly allocated to receive either a removable air boot (intervention) or plaster cast (control) at the 2 weeks postsurgery appointment. All participants were provided a group-specific exercise leaflet demonstrated by a research physiotherapist. Participants were encouraged to perform exercises as often as pain allowed, advised on gait re-education with crutches and progress to weight bearing as able. Removable support boots could be any make or model with at least two air cells, rigid (eg, plastic) anterior and posterior outer sections, and were suitable for weight bearing. Casts were standard below-knee rigid shells with soft lining which supported the ankle in plantigrade and were provided with a plaster shoe to allow weight bearing. At the 6-week appointment, participants had their cast or boot removed (as appropriate) and were followed up for an additional 6 weeks, by which time they were expected to return to normal activities.

## Overview of economic evaluation

We conducted a cost-utility analysis and a cost-effectiveness analysis, comparing the boot with the cast group at 12 weeks after ankle stabilisation surgery. The primary analysis took a National Health Service plus Personal Social Services (NHS+PSS) perspective. The secondary analysis took a societal perspective on costs, which included private expenses, informal care and productivity losses. All analyses followed the prespecified health economics analysis plan[15] and were performed in STATA version 17. The CHEERS 2022 checklist was followed for the reporting the results of the economic evaluations.

## Resource use data collection for the economic evaluation

At 2 weeks postsurgery, the physiotherapist and/or plaster technicians recorded staff grade, time and resources required to fit the boot or the plaster, including brand of boot. Research nurses/physiotherapists reviewed medical notes at trial centres to collect data on further hospital visits within the study 10-week period. Resources used within the first 2 weeks of surgery were not collected, as randomisation took place 2 weeks after surgery. Other resource use was collected from participants in postal questionnaires at 6 and 12 weeks follow-up points. Questionnaires asked about physiotherapy received in hospital or in the community; general practice contacts with doctors and nurses; additional outpatient appointments or accident and emergency (A&E) attendances and admissions to hospitals outside the recruiting centre; NHS equipment (eg, walking aids); prescribed medications; contacts with social worker, home care help services and food-at-home services; privately paid physiotherapy, medical equipment and home care or food services; productivity losses measured in terms of time-off work (absenteeism) and unproductive time at work (presenteeism); and time spent on informal care by a friend or relative.

## Outcome data collection for the economic evaluation

Outcome data were collected at baseline, 6, 7 and 12 weeks postsurgery. We used responses to the EQ-5D-5L questionnaire to compute a utility score to estimate quality-adjusted life years (QALYs).[16] The EQ-5D-5L is a patient-reported outcome instrument, standardised and validated to measure generic health-related quality of life.[16] It includes five dimensions: mobility, self-care, usual activities, pain/discomfort and anxiety/depression. It is widely used in economic analysis in clinical trials as it allows for the direct comparison of health benefits across different clinical interventions.

OMAS was the primary clinical outcome of the trial. The OMAS is patient-reported outcome measure with scores varying between 1 and 100. A clinically meaningful difference in OMAS was defined as 10-points or more.[17]

## Valuing resource use to derive costs

We used a microcosting approach to estimate the cost of delivering treatment in both arms. We used local procurement prices for the removable boots and plaster materials,

and valued staff grades and time spent delivering both treatments using 2020/2021 prices from the Unit Costs of Health and Social Care 2021 (PSSRU).[18] Secondary care resources were valued using the UK NHS National Collection of Costs 2019/2020[19] (2019/2020 prices were inflated to 2020/2021 prices using the NHS Cost Inflation Index for Pay & Prices). Medication costs were derived from the British National Formulary for medications (2022 prices).[20] Productivity losses were valued using a human capital approach and the ONS averaged gross earning per hour (2021) for the relevant age group of the trial participants (age 40–49),[21] and the minimum wage rates between April 2021 and March 2022 for informal care.[22] Unit costs applied and their sources are presented in the Appendix online supplemental table S1. Costs were calculated by multiplying the units of resource used by its unit cost.

## Valuing EQ-5D-5L to derive QALYs

EQ-5D-5L utility scores at baseline, 6, 7 and 12 weeks postoperative were mapped from the 3L version using Hernandez Alava and Pudney's algorithm.[23] Utility scores are bound at a maximum of 1 (corresponding to perfect health), where 0 corresponds to death. Negative values were permitted for health states worse than death. We calculated accumulated QALY gains per patient using the area-under-the-curve approach, assuming a linear change between utility scores at each time point.[24] For comparison, we also produced utility scores using Van Hout's algorithm, which was the recommended mapping algorithm by NICE before 2022.[25]

## Cost-utility analysis

Analyses were intention-to-treat, where participants' data were analysed based on their randomly allocated treatment group.

We imputed missing costs, utility scores and OMAS at each time point using multiple imputation models[26] with chained equations, with 50 sets and predictive mean matching, assuming that data were not missing completely at random (details of imputation model in 'Appendix Imputation' of missing data). Costs and outcomes were then adjusted for hospital site (trial stratification variable) and prespecified variables (age, sex and fracture complexity).[15] QALYs and OMAS were further adjusted for baseline scores.[24] We plotted costs and outcomes distributions to explore differences between the groups and cost drivers.

We jointly estimated the difference between groups in terms of costs and outcomes using seemingly unrelated regressions and computed the correlation of residuals.

We estimated cost-effectiveness in relation to QALYs from the NHS+PSS and societal perspectives using the incremental net monetary benefit (INMB) with a willingness-to-pay (WTP) threshold of £20 000 per QALY.[27] We plotted the results of 1000 bootstrapped iterations of the incremental costs and incremental effects in cost-effectiveness planes (CEP) and the corresponding

**Table 1** Participants baseline characteristics

| | | Boot (n=123) | | Cast (n=120) | |
|---|---|---|---|---|---|
| | | **Mean** | **SD** | **Mean** | **SD** |
| Age | n=243 | 48.75 | 15.71 | 47.7 | 17.1 |
| BMI | n=239 | 28.29 | 5.79 | 27.42 | 5.54 |
| | | **Count** | **%** | **Count** | **%** |
| Gender | Male | 46 | 37.4% | 49 | 40.8% |
| | Female | 77 | 62.6% | 70 | 58.3% |
| | Missing | 0 | 0.0% | 1 | 0.8% |
| Fracture complexity | Simple | 101 | 82.1% | 99 | 82.5% |
| | Other | 21 | 17.1% | 19 | 15.8% |
| | Missing | 1 | 0.8% | 2 | 1.7% |
| Living alone prior to injury | Yes | 96 | 78.0% | 95 | 79.2% |
| | No | 25 | 20.3% | 18 | 15.0% |
| | Missing | 2 | 1.6% | 7 | 5.8% |
| Marital status | Single | 34 | 27.6% | 34 | 28.3% |
| | Married | 56 | 45.5% | 61 | 50.8% |
| | Civil partnership | 8 | 6.5% | 9 | 7.5% |
| | Divorced/partnership dissolved | 16 | 13.0% | 3 | 2.5% |
| | Widowed/surviving civil partner | 6 | 4.9% | 6 | 5.0% |
| | Missing | 3 | 2.4% | 7 | 5.8% |
| Education | None | 9 | 7.3% | 15 | 12.5% |
| | GCSE and A/AS level | 49 | 39.8% | 47 | 39.2% |
| | First degree | 21 | 17.1% | 18 | 15.0% |
| | Higher degree | 19 | 15.4% | 14 | 11.7% |
| | Other | 11 | 8.9% | 13 | 10.8% |
| | Missing | 14 | 11.4% | 13 | 10.8% |
| Employment | Full-time paid employment | 62 | 50.4% | 53 | 44.2% |
| | Part-time paid employment | 15 | 12.2% | 15 | 12.5% |
| | Retired | 24 | 19.5% | 20 | 16.7% |
| | Volunteer | 0 | 0.0% | 1 | 0.8% |
| | Unemployed | 4 | 3.3% | 4 | 3.3% |
| | Looking after home | 7 | 5.7% | 7 | 5.8% |
| | Full-time education | 2 | 1.6% | 6 | 5.0% |
| | Other | 7 | 5.7% | 7 | 5.8% |
| | Missing | 2 | 1.7% | 7 | 5.8% |

A/AS, advanced level or advanced subsidiary level; BMI, body mass index; GCSE, General Certificate of Secondary Education.

cost-effectiveness acceptability curves (CEAC) to illustrate the uncertainty surrounding the decision to adopt the intervention.

### Uncertainty and sensitivity analyses

We conducted a complete case analysis to illustrate the impact of missing data and the imputation model in results. To account for model uncertainty, we adjusted costs and QALYs also including relationship status, alone living status, employment status, education and body mass index. We carried out one-way sensitivity analysis for scenarios where we decreased the price of boots by 25%, 50% and 75% to explore the impact of lower prices of boots on the results.

### Overview of the qualitative study

Participants were invited to participate in a qualitative interview towards the end of their study period, at approximately 12 weeks postsurgery. Two qualitative researchers chose individuals from the pool of positive responders, ensuring a balance of treatment options, sex, age, hospital site and baseline OMAS scores. Participants were invited

**Table 2** Costs and outcomes by trial arm and by perspective on costs (imputed data)

| | Boot (N=123) | Cast (N=120) | Difference |
|---|---|---|---|
| NHS+PSS | Mean (95% CI) | Mean (95% CI) | Mean (95% CI) |
| Ankle treatment (boot or cast) | £203 (£200 to £206) | £179 (£177 to £181) | £24 (£20 to £28) |
| Physiotherapy in the community | £11 (£4 to £18) | £4 (£1 to £7) | £7 (–£1 to £14) |
| Physiotherapy in the hospital | £26 (£15 to £37) | £24 (£12 to £36) | £2 (–£14 to £18) |
| GP practice services | £57 (£30 to £83) | £37 (£9 to £64) | £20 (–£18 to £58) |
| Social care services | £19 (£8 to £30) | £6 (£0 to £12) | £13 (£0 to £25) |
| A&E and outpatient services | £55 (£25 to £85) | £31 (£18 to £43) | £24 (–£9 to £57) |
| Hospital admissions | £8 (–£7 to £23) | £8 (–£7 to £24) | –£0 (–£22 to £22) |
| NHS equipment | £3 (£1 to £5) | £3 (£1 to £5) | £0 (–£2 to £3) |
| Medications | £7 (£4 to £9) | £5 (£1 to £8) | £2 (–£2 to £6) |
| Private costs and productivity losses | | | |
| Private physiotherapy | £16 (£5 to £27) | £19 (£7 to £31) | –£3 (–£19 to £13) |
| Additional medical equipment | £5 (£2 to £9) | £1 (–£0 to £2) | £4 (£0 to £8) |
| Private community care | £0 (–£0 to £1) | £1 (–£1 to £4) | –£1 (–£3 to £2) |
| Other private expenditures on services/activities | £105 (£75 to £134) | £139 (£31 to £247) | –£34 (–£145 to £76) |
| Other major expenditure not reported elsewhere | £10 (–£1 to £20) | £20 (£2 to £38) | –£10 (–£31 to £10) |
| Productivity loss (time-off work and leisure) | £2359 (£1912 to £2807) | £2995 (£2334 to £3657) | –£636 (–£1427 to £155) |
| Informal care | £888 (£567 to £1209) | £1025 (£630 to £1420) | –£137 (–£642 to £367) |
| Outcomes | | | |
| QALYs gained (Hernandez Alava and Pudney) | 0.1195 (0.1144 to 0.1247) | 0.1248 (0.1208 to 0.1287) | –0.0052 (–0.0117 to 0.0013) |
| QALYs gained (Van Hout) | 0.1117 (0.1062 to 0.1172) | 0.1189 (0.1146 to 0.1231) | –0.0071 (–0.0141 to 0.0002) |
| Olerud and Molander (primary outcome, week 7-week 2 baseline) | –55.0 (–59.1 to 51.0) | –59.0 (–62.3 to 55.7) | 4.0 (–1.2 to 9.2) |
| Olerud and Molander (week 12-week 2 baseline) | –39.7 (–43.7 to 35.8) | –35.5 (–39.1 to 31.8) | –4.3 (–9.7 to 1.1) |

Costs are in 2020/2021 prices.
A&E, accident and emergency; GP, general practice; NHS+PSS, National Health Service and Personal Social Services; QALYs, quality-adjusted life years.

to a 45–60 min telephone semistructured interview; a date and time to suit the participants was set for those who agreed to participate. Data were audio recorded and transcribed verbatim. During the interviews, participants were asked to discuss their experiences of living with the treatment (boot or cast) as well as exploring the impact this had on their activities of daily living (personal hygiene, sleep, mobility, work, diet), psychological implications (mood and independence) and wider impact on their family and social life (changes to routines, caring responsibilities, finances).

## Analysis of qualitative data

Qualitative interviews were analysed using the six-step process of thematic analysis by Braun and Clarke.[28] We read transcripts to gain a sense of the whole before step-by-step analysis of each transcript to identify codes, categories and initial themes. Once all interviews were analysed individually, a cross case analysis was undertaken, identifying themes across the whole dataset. These themes and the process of analysis were shared with a second qualitative researcher to ensure credibility of the analytical process.

## RESULTS

### Sample

243 participants were randomised between August 2015 and September 2018 (n=123 boot group and n=120 plaster group). Table 1 describes participant demographics and socioeconomic characteristics by arm. No major differences were observed between trial arms for all variables.

Sixteen trial participants recruited across the eight study sites consented to be interviewed; six were men and 10 were women, between the ages of 24 and 77 years (mean age was 49.2, median age was 50.5); seven received a plaster cast and nine received a boot. Most ankle fractures were simple (n=14), while two were comminuted

**Table 3** Cost-utility analysis results

| Cost-utility analysis | NHS+PSS perspective Mean (95% CI) | Societal perspective Mean (95% CI) |
|---|---|---|
| **Base case** | | |
| Incremental cost | £88 (£22 to £155) | −£676 (−£1689 to £337) |
| Incremental QALY | −0.0020 (−0.0067 to 0.0026) | −0.0021 (−0.0067 to 0.0026) |
| Net monetary benefit | −£129 (−£231 to −£28) | £634 (−£266 to £1535) |
| Probability boot is cost-effective at £20 000/QALY | 1.3% | 88.6% |
| **Sensitivity analysis** | | |
| **Models type 2—adjusting for additional covariates** | | |
| Incremental cost | £74 (£8 to £140) | −£1017 (−£1967 to −£67) |
| Incremental QALY | −0.0021 (−0.0067 to 0.0025) | −0.0021 (−0.0067 to 0.0025) |
| Net monetary benefit | −£116 (−£217 to −£15) | £974 (£99 to £1850) |
| Probability boot is cost-effective at £20 000/QALY | 2.3% | 97.5% |
| **Complete case analysis** | | |
| Incremental cost | £123 (£19 to £227) | £323 (−£934 to £1580) |
| Incremental QALY | −0.0010 (−0.0076 to 0.0056) | −0.0027 (−0.0096 to 0.0041) |
| Net monetary benefit | −£143 (−£327 to £40) | −£378 (−£1771 to £1015) |
| Probability boot is cost-effective at £20 000/QALY | 4.1% | 28.3% |

Models type 1 (base case) adjust for age, gender, fracture complexity and hospital site; models type 2 also include relationship status (married/partnership vs other), alone living status, employment status (working part or full time vs other), education (higher degree vs other) and BMI.
INMB measured at a willingness to pay for a QALY of £20 000.
INMB, incremental net monetary benefit; NHS+PSS, National Health Service and Personal Social Services; QALY, quality-adjusted life year.

(bone is broken into more than two pieces) and there was a range of baseline OMAS scores between 5 and 80.

## Costs and outcomes

Table 2 reports costs and outcomes by trial arm including imputed data. Units of resource use per arm is presented in the online supplemental appendix table S2 and all available cost data (complete cost categories only) are presented in online supplemental appendix table S3. The ankle management treatment cost difference reflects the difference in price between the boot (£50) and plaster (£25), and the difference in staff grade and time fitting each. On average, it took the technicians 25.3 min to fit the cast (SD 17.1) and 23.7 min to fit the boot (SD 24.9).

NHS+PSS cost differences are small and in favour of the plaster group. From a societal perspective, the differences are larger, particularly in relation to productivity loss and informal care, and favour the boot group (online supplemental appendix figures S1 and S2 of the Appendix).

## Cost-effectiveness base case and sensitivity analyses

Table 3 presents the cost-effectiveness analysis results from an NHS+PSS perspective and a societal perspective for the base case and sensitivity analyses. From an NHS+PSS perspective, treatment in the boot group costs, on average, £88 (95% CI £22 to £155) per patient more than in the plaster group. From a societal perspective, the boot saved, on average, £676 per patient compared with plaster (95% CI −1689 to £337). The cost savings for the

boot are driven by reduced informal care and time-off productive work. There was no evidence of a QALY difference between the arms (mean QALY gain −0.0020; 95% CI −0.0067 to 0.0026).

From an NHS+PSS perspective, the boot is dominated, with a negative INMB statistic (mean −£129 (95% CI −£231 to −£28)). When including additional privately incurred costs, productivity losses and the burden of informal care in a societal perspective, the boot is cost-effective when willing-to-pay £20 000 per QALY, with a mean INMB £634 (95% CI −£266 to £1535). The societal perspective results, although deriving larger net benefits, are also more uncertain, with wider confidence intervals that cross the null.

The CEP for the NHS+PSS perspective shows a high concentration of results in the upper left quadrant (online supplemental figure 3A of the Appendix) reflecting a dominated treatment option. This corresponds to a low probability of the boot being cost-effective even at high WTP thresholds in the CEAC (online supplemental figure 4A of the Appendix). From a societal perspective (online supplemental figure 3B of the Appendix) shows a high concentration of results in the lower-left quadrant, but a higher likelihood of being cost-effective (online supplemental figure 4B of the Appendix).

These findings are robust in sensitivity analyses to the model specification. Our findings are sensitive to the imputation model. In a complete case analysis, the

**Table 4** Quotes from the qualitative interviews

| Topic | Quotes from the qualitative interviews |
|---|---|
| Feelings related to ankle fracture | *"I had depression before and I think that didn't help that I've got that tendency as well…there was some sense of isolation more to do with not being able to see people a lot of the time or not having the same routine. The other thing I found really challenging was having to ask for help"* (Female 44 cast)<br>*"It had a huge impact on my independence because it meant I was relying on other people to even leave the house"* (Female 52 cast) |
| Daily activities | *"There's only so much you want to be waited on, everyone says I dream of that but when it comes to it, it actually drives you insane, to be able to get back and cook meals for my children and to be able to go and make myself a cup of tea and then carry it to another room is fantastic* (Female 46 boot)<br>*"…for me to be reliant on somebody else, it was degrading…and I felt vulnerable like in a really vulnerable state definitely …I found [with boot] I was less dependent on other people because I could move around a bit more, I could have a bath or shower myself, I could get in and out of bed on my own"* (Female 27 boot)<br>*"It put a huge burden on my daughter who was having to come 4 times a day, but then [with boot] I could go into the kitchen for a large degree to, make a meal for myself you know"* (Female 77 boot) |
| Social life and events | *"Basically, I was stuck around the house, we had to cancel our holiday… to be honest I just sat round the house pretty much for six weeks"* (Male 69 cast)<br>*"there were family BBQs where basically my husband took the kids and I just relaxed at home, it was a bit too much"* (Female 44 cast)<br>*"I feel the boot is a very good protection around my leg and ankle particularly if I was out and I was using the wheelchair quite a lot as we have a busy social life… I think would have been more twitchy had I had a plaster"* (Female 69 boot)<br>*"We hired a wheelchair for me because we went to the [event] with the kids in [place] and that was a whole day come night-time thing"* (Female 46 boot) |
| Private expenses | *"I obviously had to buy a waterproof cover for the leg…. I went out and brought a couple of pairs of cropped leggings…I was obviously very limited what I could wear because of the plaster, probably spent about £30 on the cover and leggings"* (Female 52 cast)<br>*"I could still wear leggings, I didn't have to cut up any trousers"* (Female 27 boot)<br>*"I bought one of those like big hot water bottle kind of thing that went up your leg for showers and things which costs about £14"* (Female 49 cast)<br>*"I had to cut up 4 pairs of my leggings up because my leggings wouldn't go over the plaster"* (Female 54 cast) |
| Mobility | *"…because the foot got really swollen (from toes to my knee), so the advantage of taking the boot off and putting peas on it, in your own mind you were doing something to ease the cause"* (Female 53 boot)<br>*"One of the main advantages was I was able to start my physio a lot earlier… I was able to practice flexing and getting my foot up and turning it. The flexibility in my foot its better, it's completely back to normal now where I think if it was in a cast… everything would have started to seize up a lot more and would have been harder to get the flexibility back"* (Female 46 boot). |
| Quality of life | *"I found the boot easier than the plaster. The plaster was very heavy and I found with the boot I could actually weight bear a bit more and get on with day to day running of life"* (Female 53 boot)<br>*"I wanted the boot because I knew the boot would be better for me personally… because I am so active I wanted to get around and do things"* (Male 24 boot) |

societal perspective results are closer to those of the NHS+PSS results, with the boot group reporting, on average, higher costs than the cast group. When boot prices decreased by 25%, 50% and 75% in sensitivity analyses, the NHS+PSS cost differences between arms decreased to £76 (95% CI £9 to £142), £63 (95% CI –£3 to £130) and £51 (95% CI –£16 to £117), respectively, indicating no evidence of a differences in costs, QALYs or INMB statistics (see Appendix online supplemental appendix table S4).

### Qualitative findings

The economic results of lower productivity losses and informal care costs in the boot group were reflected in the qualitative interviews. Having a fracture irrespective of the treatment offered (boot or cast) increased reliance on others and this was typically managed by family members and friends which was frustrating for participants, many of whom expressed feeling helpless (table 4—feelings).

Those with the boot expressed an ability to become either more self-caring or take back family responsibilities earlier, such as walking to the kitchen to prepare dinner and making their own drinks. These activities were harder for those in cast due to immobility (table 4—daily activities).

It was also apparent that having a boot enabled participants to continue their wider social connections; they travelled further and continued with wider social engagements (outside the home) than those with a cast. In contrast, those with a cast talked about having to miss family engagements and wider social activities where

people with the boot tended to continue to attend these, even if it required hiring additional support, such as a wheelchair (table 4—social life).

Patients in the cast group appeared to have increased personal costs purchasing additional clothing and equipment to accommodate the use of the cast (table 4— private expenses).

Patients in the cast group expressed difficulty with stair climbing, while those with boots who described being able to weight bear did not express similar difficulties. Another advantage identified by participants in boots was the ability to remove it to undertake the prescribed exercises, which they felt aided a quicker recovery (table 4— mobility). Of the five participants who expressed a preference, two in cast would have preferred boots and the other three were glad they had received boots.

In the qualitative interviews, participants reported that they felt the boot enhanced their quality of life (table 4— quality of life).

## DISCUSSION
### Statement of principal findings
We found that the boot group had, on average, higher NHS+PSS costs but differences between groups were small. Differences in QALY gains between groups were negligible. When including costs with informal care burden and productivity losses from time-off or unproductive work, the boot saves society on average £676 and could be a cost-effective intervention when willing-to-pay £20 000 per QALY. Qualitative interviews with 16 participants complemented the quantitative analysis to better understand patients' perspectives. Lower productivity losses and informal care need in the boot group were reflected in the interviews, and some participants in the boot group highlighted that they were more satisfied with their treatment option, felt more independent and were able to return to their usual activities sooner.

### Strengths and limitations
Our study reports a full economic evaluation from an NHS+PSS perspective and a nested qualitative study conducted alongside a large pragmatic multicentre RCT in the management of ankle factures after surgery. The nested qualitative study shed further light onto the interpretation of economic findings. The ART trial is the largest RCT to date comparing removable support boot with a plaster cast for early mobilisation after ankle fracture surgery and ours is the first cost-effectiveness evaluation comparing the two ankle management treatment options.

Despite data collection taking place several years ago, we believe our results are relevant and transferable to current NHS practice. The ART trial was powered to detect a difference in the primary outcome (OMAS score at 7 weeks), and not difference in costs, QALYs or INMB statistics, consequently there is more uncertainty in the results, especially from the societal perspective. Missing data for the economic evaluation was relatively high and not completely at random. Our base-case results include imputed data, due to the risk of biases in a complete case analysis, which changed the results for the societal perspective. The larger cost driver in favour of the boot is informal care burden. Measuring the burden for carers in economic evaluation is often neglected and methods to value carer burden are not contemplated in the NICE reference case.[27]

The EQ-5D-5L questionnaire appears to be too insensitive to pick up important aspects of quality of life that the boot can provide to patients. It does not directly measure the broader aspects of well-being and ability that were captured in the qualitative interviews. The perception of increased well-being in the qualitative interviews, however, may be a result of expectations towards the newer technology, which is difficult to objectively measure.

### Results in context
A recent systematic review of the clinical effectiveness and cost-effectiveness of orthotic walking boots for patients with ankle fractures or ligament injuries found no previous cost-effectiveness studies comparing boots with casts.[12] According to our knowledge, this is the first cost-effectiveness evaluation alongside a trial comparing boots with cast.

Nevertheless, a few other studies have also compared resource use, return to work or to normal activities between the functional treatment and cast groups. For example, Egol *et al* reported earlier return to work in the brace group for patients who underwent internal fixation[29]; however, these patients were all instructed to avoid weight bearing on the affected side for 6 weeks. Honignmann found that for patients who had malleolar fractures followed by ORIF, patients in the orthesis group with prescribed full weight bearing returned to work sooner; however, the difference was not significant.[30] Simansky *et al* and Lehtonen *et al* no significant differences in return to work comparing functional treatment versus cast patients who underwent ORIF.[11 31]

In our study, no differences were observed in how soon participants returned to driving or full preinjury work duties. The boot group indicated at the 6-week postoperative time point that their injury was having a greater impact on daily activities; however, this difference was no longer present 4 weeks later.

Similarly to our results, Kearney *et al* and Haque *et al* found no significant difference between brace and cast groups in terms of EQ 5D-5L scores at any time point.[32 33]

### Conclusion
While the boot is more expensive than cast for the NHS+PSS payer, it reduces productivity losses and the need for informal care and empowers patients. Given that differences in QALYs and costs to the NHS are small, the decision to use a boot or cast following ankle surgery could be left to patients' and clinicians' preferences.

**Author affiliations**

¹Musculoskeletal Research Unit, Bristol Medical School, Translational Health Sciences, University of Bristol, Bristol, UK
²UKPRP VISION Consortium, London, UK
³School of Nursing and Society, University of Salford, Salford, UK
⁴Department of Social Science, University of Stavanger, Stavanger, Norway
⁵University Hospitals Dorset NHS Foundation Trust, Poole, UK
⁶Department of Medical Science and Public Health, Bournemouth University, Poole, UK
⁷Bournemouth University Clinical Research Unit, Bournemouth University, Poole, UK
⁸Peninsula Clinical Trials Unit, Plymouth University, Plymouth, UK
⁹Exeter Clinical Trials Unit, University of Exeter, Exeter, UK
¹⁰NIHR Biomedical Research Centre, University of Bristol, Bristol, UK

**Acknowledgements** The views expressed herein are those of the authors and not necessarily those of the NHS, the NIHR or the Department of Health and Social Care. We thank Pete Thomas for his immense contribution in designing the study, securing study funding and in writing the statistical analysis plan that had influenced the health economics analysis plan.

**Contributors** RM, EMRM, CH and HA conceived the idea for the study and secured funding. EMRM designed the economic evaluation and supervised PB and ECB in performing the economic evaluation. PB, ECB and EMRM had access to data and conducted the economic analysis. VH designed and VH and HA conducted the qualitative study. LT was trial manager with oversight of data collection and management. SD managed the data sources and performed the analysis of clinical outcomes. ECB, PB, EMRM and VH drafted the manuscript. All authors critically revised the manuscript and approved the final version for submission. EMRM and BS act as guarantors.

**Funding** This report is independent research funded by the National Institute for Health and Care Research (NIHR) under its Research for Patient Benefit Programme 'Does early mobilisation after Ankle fracture surgery enhance Recovery? A pragmatic multi-centre randomised controlled Trial with qualitative component and health economic analysis comparing the use of plaster versus support boot (ART)', PB-PG-0213-30021. The views expressed in this publication are those of the author(s) and not necessarily those of the NHS, the National Institute for Health and Care Research or the Department of Health and Social Care. ECB's salary is supported by the UK Prevention Research Partnership (Violence, Health and Society; MR-VO49879/1), a Consortium funded by the British Heart Foundation, Chief Scientist Office of the Scottish Government Health and Social Care Directorates, Engineering and Physical Sciences Research Council, Economic and Social Research Council, Health and Social Care Research and Development Division (Welsh Government), Medical Research Council, National Institute for Health and Care Research, Natural Environment Research Council, Public Health Agency (Northern Ireland), The Health Foundation, and Wellcome. The views expressed are those of the researchers and not necessarily those of the UK Prevention Research Partnership or any other funder.

**Competing interests** None declared.

**Patient and public involvement** Patients and/or the public were involved in the design, or conduct, or reporting, or dissemination plans of this research. Refer to the Methods section for further details.

**Patient consent for publication** Not applicable.

**Ethics approval** This study involves human participants and was approved by South Central – Hampshire A Research Ethics Committee (reference 14/SC/1409). Participants gave informed consent to participate in the study before taking part.

**Provenance and peer review** Not commissioned; externally peer reviewed.

**Data availability statement** Data are available upon reasonable request. The datasets used and/or analysed during the current study are available from the corresponding author on reasonable request.

**ORCID iDs**
Petra Baji http://orcid.org/0000-0003-2899-8557
Estela C Barbosa http://orcid.org/0000-0001-8282-131X
Vanessa Heaslip http://orcid.org/0000-0003-2037-4002
Lee Tbaily http://orcid.org/0000-0003-4203-480X
Sharon Docherty http://orcid.org/0000-0003-3209-1560
Christopher Hayward http://orcid.org/0000-0002-2666-5073
Elsa M R Marques http://orcid.org/0000-0003-1360-5677

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
