## [Reviewer comments · BMJ Open]

ARTICLE DETAILS

TITLE (PROVISIONAL)	Use of removable support boot vs cast for early mobilisation after ankle fracture surgery: Cost-effectiveness analysis and qualitative findings of the Ankle Recovery Trial. (ART)
AUTHORS	Baji, Petra; Barbosa, Estela; Heaslip, Vanessa; Sangar, Bob; Tbaily, Lee; Martin, Rachel; Docherty, Sharon; Allen, Helen; Hayward, Christopher; Marques, Elsa

VERSION 1 – REVIEW

REVIEWER	Iwona Bielska McMaster University, Department of Health Research Methods, Evidence, and Impact
REVIEW RETURNED	23-May-2023

GENERAL COMMENTS	Thank you for giving me the opportunity to review this manuscript, which describes the results of a trial examining two types of immobilization following ankle fracture surgery. This is an important area of study and there is limited information on the cost-effectiveness of various interventions. The authors supplement the quantitative information with qualitative data collected from a subset of participants, which provides a patient perspective on the use and type of immobilization. In general, it would be very helpful if the authors included more information on the trial in the Methods section. Although the detailed protocol for the study is included in the Appendix, it is imperative that the paper summarizes the trial so that the readers who only read the manuscript have an overview of what was done. For example, information on study setting, participants (including inclusion/exclusion criteria) are needed. Also, the authors should describe the clinical results of the trial early on in the manuscript and indicate how that paper differs from the current paper (i.e., reporting of clinical versus cost-effectiveness results). Please describe the sample size in more detail as it is referred to elsewhere (it is indicated that the trial was not powered to detect differences in costs, QALYs). If the trial is not powered to detect differences in costs, how does this affect the cost-effectiveness analyses? Why are the results of the trial which was conducted in 2015-18 being reported on now? This is perhaps not necessary to include in the manuscript itself, however, it is important to reflect on if there have been any practice or intervention changes since 2015-18 that would affect the relevance of the results.
--

	Please describe why participants were followed up to 12-weeks (for example, around line 28 in the Background). In the Background, please include the rationale for the trial, summaries of any other cost-effectiveness studies of immobilization interventions following ankle fracture surgery, and an overview of the literature on the economic impact of ankle fractures. Furthermore, information on the trial registration and the CONSORT checklist should be included. Page 7, line 9 - perhaps reword the sentence as it seems to indicate that reference 10 is written by the authors. For the economic information, please indicate the year and currency of the results. Also, include a sentence on why discounting was not used. If the study was conducted in 2015-18, why are 2021-22 wage rates used? Were the costs adjusted for inflation and if so, how was this done? For the qualitative study (page 10, lines 35-42), what was the response rate? What was the mean and median age of participants? Discussion - page 15, line 9 - first sentence may need rewording "The found...". Lines 16-20 - the qualitative results may be overstated as it is difficult to compare personal opinions. The sample size for the qualitative study should be indicated here again (n=...). Results in context - page 16 - please provide more information on the studies that have been conducted to date on this topic (e.g., sample sizes, locations, limitations) and how they compare to your results. Page 16, line 13 - please cite the paper on the clinical results from the trial. Please also include more information in the Background and Discussion on the clinical results that were found in the trial. Page 19 - Patient consent form - it states in this section that no person's data are included in the manuscript, however, the manuscript involves quotes from participants in the qualitative study with specific information provided (e.g., participant who mentions MTV Clashes in Plymouth). Table S1 - page 31 - please include year of costs. For hospital procurement, when were the data obtained (e.g., year)?
--	---

REVIEWER	Kentaro Amaha St Luke's International University
REVIEW RETURNED	27-May-2023

GENERAL COMMENTS	It is a well written manuscript with originality and comprehensible review of the literature. We believe that this is an interesting topic for foot and ankle surgeons and that this knowledge is useful for clinical practice. It could be beneficial for the scientific community as it is a prospective study with clear message. More specific comments are the following:
--

	TITLE No comment. ABSTRACT Clear and comprehensive. INTRODUCTION In the introduction section, there has been done a meticulous work to provide a short but adequate background of the study. Besides, the importance of the study is also designated through this section and the purpose is also clearly cited. METHODS The research methodology is detailed and well written. RESULTS The outcomes of the study are presented comprehensively. However, the results section is long, please make it more concise. DISCUSSION In the DISCUSSION, the content is concise and well written. Additionally, the clinical relevance of the study is well presented and the limitations of the study are plainly analyzed. REFERENCES No comment. FIGURES No comment. TABLES No comment. While Cost-effectiveness analysis is a great methodology, the results (estimates) obtained from it are quite unstable and can easily change depending on assumptions. I am not that familiar with statistics so we should check with a statistician to see if there are any problems.
--	---

VERSION 1 – AUTHOR RESPONSE

Reviewer: 1

Dr. Iwona Bielska, McMaster University, Jagiellonian University

Comments to the Author:

Thank you for giving me the opportunity to review this manuscript, which describes the results of a trial examining two types of immobilization following ankle fracture surgery. This is an important area of study and there is limited information on the cost-effectiveness of various interventions. The authors supplement the quantitative information with qualitative data collected from a subset of

participants, which provides a patient perspective on the use and type of immobilization.

We thank the reviewer for the positive feedback on the paper.

R.1.1. In general, it would be very helpful if the authors included more information on the trial in the Methods section. Although the detailed protocol for the study is included in the Appendix, it is imperative that the paper summarizes the trial so that the readers who only read the manuscript have an overview of what was done. For example, information on study setting, participants (including inclusion/exclusion criteria) are needed.

Thank you for this suggestion. We have added more information of the trial in the Methods section, e.g. inclusion and exclusion criteria, registration number.

'A health economic evaluation and qualitative component were included in the ART pragmatic multi-centre RCT to provide a more inclusive picture. The trial was conducted across eight UK NHS secondary care trusts between July 2015 and November 2018 and compared casts with removable support boots, combined with weight bearing as able, as methods of ankle fracture management from two weeks post-surgery. Most participants at the 6-week follow-up timepoint were expected to remove their plaster cast or boot and start to mobilise freely. Questionnaires were completed at randomization (2 weeks after surgery), then at 6-, 7-, and 12-weeks post-surgery. The primary clinical endpoint was the Olerud and Molander ankle score (OMAS) 14 measured at 7-weeks post-surgery, when participants should have adapted to mobilising without plaster cast or boot. The ART trial was approved by the South Central – Hampshire A Research Ethics Committee (reference 14/SC/1409) and was prospectively registered (ISRCTN: 15497399)...' (See in Methods/ Trial design and ethics approval)

'All adults (>16 years old) undergoing open reduction and internal fixation (ORIF) surgery for an unstable ankle fracture were eligible for the study. Exclusion criteria were: open ankle fracture; concern about quality of fixation/wound integrity; requiring further stabilisation in/around the ankle (e.g. syndesmosis); active leg ulceration; poor skin condition at operating site; serious concomitant disease, diabetic/other sensory neuropathy; non-ambulatory prior to injury; inability to understand/complete outcome questionnaires; enrolment in other interventional research which may confound data collection; concomitant injuries which may affect rehabilitation.' (See in Methods/ Participants)

R.1.2. Also, the authors should describe the clinical results of the trial early on in the manuscript and indicate how that paper differs from the current paper (i.e., reporting of clinical versus cost-effectiveness results).

The clinical paper will report the results on the primary outcome measure of the trial, the Olerud and Molander Ankle Symptom Score (OMAS) at week 7, and on secondary outcomes, such as OMAS scores at week 12, EQ-5D VAS scores at week 7 and 12, secondary mechanistic measures such as ankle ROM (dorsiflexion, plantarflexion, inversion and eversion), calf circumference, and ankle swelling at 6 weeks, as well as weight-bearing status, use of walking aids, return to driving and work, impact on daily activities at weeks 6 and 12. Complications and serious adverse events are also reported at 6 and 12 weeks in the clinical paper. These can be also found in the Final report to the Funder which is available from the authors upon request.

We now fully specify the trial outcomes in page 7 of the methods:

'Results on clinical effectiveness (such as primary outcome measure and secondary outcomes, such as OMAS at week 12, EQ-5D VAS scores at week 7 and 12, secondary mechanistic measures at 6 weeks, weight-bearing status, use of walking aids, return to driving and work, impact on daily activities at weeks 6 and 12, as well as complications and serious adverse events at 6 and 12 weeks) can be found in the final report to the Funder ¹³ and will be reported in a separate paper.' (See Methods/ Trial design and ethics approval)

This cost-effectiveness paper results on costs from the NHS+PPS and social perspective, on QALY-s, and on incremental net monetary benefit statistics (INMB) measured 12-weeks after surgery, for a society willing-to-pay £20,000 per QALY.

We have also added the following summary of the main results of the clinical paper in the Background section:

'Clinically there were no observed differences in ankle function between treatment groups at any time point. Overall early weight bearing in both the boot or cast was safe with low wound complication rates (7%) that, although more frequent in the boot group, were minor.' (See Background section, 4th paragraph)

R1.3. Please describe the sample size in more detail as it is referred to elsewhere (it is indicated that the trial was not powered to detect differences in costs, QALYs). If the trial is not powered to detect differences in costs, how does this affect the cost-effectiveness analyses?

The study was powered to the primary outcome, the OMAS at 7 weeks post-surgery as the primary outcome measure based on an unpaired t-test with two-sided significance level (α) of 0.05 and 90% power (β), standard deviation of 21.9 and mean between group difference of 10 points on the OMAS (minimum clinically important difference). To achieve this, a total sample size of 204 (102 per group) was required, The study aimed to over-recruit by 20% to accommodate non-responders and missing data not to contribute data to the main analysis, giving a recruitment target of 246.

RCTs are usually not powered to detect differences in costs. As per Briggs 2000 [1]: Given the skewed nature of cost data and consequently higher variance of cost variables than of clinical outcomes as well as the heterogeneity of costs, the comparisons of treatment cost as well as cost-effectiveness ratios would require greater sample sizes than the corresponding clinical comparison [1]. If RCTs would be powered to the economic outcomes, they would most likely be overpowered with respect to the clinical outcomes. This might be problematic from the ethical point of view as it would probably be inappropriate to continue a trial beyond the point at which clinical superiority has been determined [1].

Not powering the trial for economic outcomes mean that we might not detect significant differences, costs, QALYs, or the economic result (incremental net monetary benefits) and our estimates would result in larger confidence intervals around cost-effectiveness results. Thus, we have to be careful when interpreting such results as we cannot be sure if differences do not exist or just not detected by the study. In this evaluation, the differences we found in QALYs are so minor, that we do not expect any meaningful difference even if we increased sample size. Difference in costs was significant from the NHS+PPS perspective, and we do not expect our conclusions to change with increasing sample size. It may be possible that increasing sample size would render the smaller difference in costs from the

societal perspective significant, by narrowing the confidence intervals of INMB results, making the boots the more cost-effective option.

We added some of this explanation to the Limitation section:

'The ART trial was powered to detect a difference in the primary outcome (OMAS score at 7 weeks), and not difference in costs, QALYs, or INMB statistics, consequently there is more uncertainty in the economic results, especially from the societal perspective.' (See in Methods/ Trial design and ethics approval)

R.1.4. Why are the results of the trial which was conducted in 2015-18 being reported on now? This is perhaps not necessary to include in the manuscript itself, however, it is important to reflect on if there have been any practice or intervention changes since 2015-18 that would affect the relevance of the results.

We thank the reviewer for raising this point. There was indeed some delay in publishing the results of this trial due to career breaks (maternity and sick leaves and the COVID pandemic). The final report of the randomised controlled trial was submitted to the Funder on 18 March 2022, after the official end date of the trial, 30 April 2019. Although there may have been changes in the surgical treatment of ankle fractures after the results of the AIM trial [2], management of post-surgical treatment has not changed noticeably and our study and its findings are still relevant today.

We agree with the reviewer that this is a limitation that needs to be mentioned, thus we added the following sentence to the Limitations section:

'Despite data collection taking place several years ago, we believe our results are relevant and transferable to current NHS practice.' (See Discussion/Limitation)

R.1.5. Please describe why participants were followed up to 12-weeks (for example, around line 28 in the Background).

At the six-week appointment participants had their cast or boot removed, and their recovery (without the boots and cast) were followed up for 6 weeks, by the time they were expected to return to their normal activities. This corresponds with finding of a previous study on the economic burden of ankle fractures, where patients returned to work after 11.2 weeks on average [3]. We have added this explanation to Method section:

'At the six-week appointment participants had their cast or boot removed (as appropriate), and were followed up for an additional 6 weeks, by which time they were expected to return to normal activities.' (See in Methods/ Intervention and usual care)

R.1.6. In the Background, please include the rationale for the trial, summaries of any other cost-effectiveness studies of immobilization interventions following ankle fracture surgery, and an overview of the literature on the economic impact of ankle fractures.

We thank the reviewer for raising this point. According to our knowledge, there are no other cost-effectiveness studies have been published so far that compares the boot and cast for ankle fracture. We added to the text the following:

'Systematic reviews conclude that well-designed, prospective studies are still needed to determine which immobilization technique offers the most benefit for treating ankle fractures. ⁷ ¹² . Furthermore, evidence on cost-effectiveness of boots compared to casts is missing from the literature. ¹² ' (See in Background 3rd paragraph)

We added some sentences on the economic impact on ankle fractures to the Background section as well:

'According to a recent systematic review, the direct costs of ankle fractures ranges from \$1,908 to \$19,555 worldwide. ⁵ In England, the cost of inpatient hospital care for ankle fractures was estimated to exceed £63.1 million in 2016/17. ¹ Costs from the social perspective are even higher due to indirect costs associated with fractures, such as productivity loss and informal care costs, that contribute to almost half of total costs. ⁶ ' (See in Background 1st paragraph)

R.1.7. Furthermore, information on the trial registration and the CONSORT checklist should be included.

The study was approved by the Health Research Authority (Ref: 14/SC/1409) and prospectively registered (ISRCTN: 15497399). The reference is added to the text. (See in Methods/ Trial design and ethics approval)

Since we do not report the effectiveness results of the randomised controlled trial, we are using the CHEERS 2022 checklist for reporting results of an economic evaluations [4] which also applies to the reporting of economic evaluations alongside clinical trials. We added a reference to the checklist in the paper and attached the checklist as supplementary material. (See in Methods/ Overview of economic evaluation)

R.1.8. Page 7, line 9 - perhaps reword the sentence as it seems to indicate that reference 10 is written by the authors.

The sentence reads: "All analyses followed the pre-specified health economics analysis plan [10]..."

The economic analysis plan for the ART trial was indeed written by some of the authors of this paper at the beginning of the trial. Please find the reference to the HEAP below:

Thomas P, **Marques EMR, Docherty S, Sangar S**: Statistical and Health Economics Analysis Plan: ART Ankle Recovery Trial version 1.1. University of Bristol. 28th Nov 2018. Available at: <https://research-information.bris.ac.uk/en/publications/statistical-and-health-economics-analysis-plan-art-ankle-recovery> Accessed: 26/01/2023

Peter Thomas retired in 2020 and therefore is not an author on this paper.

R.1.9. For the economic information, please indicate the year and currency of the results. Also, include a sentence on why discounting was not used. If the study was conducted in 2015-18, why are 2021-22 wage rates used? Were the costs adjusted for inflation and if so, how was this done?

Detailed unit costs (including currency and years for all item) are presented in Table S1 in the Appendix. We agree with the reviewer that some of these data should be added to the main paper as well. We present the results in 2020/21 prices in GBP. Primary care services, physiotherapy and community based services were valued using the unit cost database of

health and social care professionals 2020/21 (i.e, 2020/2021 prices). The latest available unit costs (National Schedule of NHS Costs 2019/20) that were available at the time of analysis were used to evaluate resource use for secondary care services: secondary care services (Accident and Emergency, Day case admission or procedure, Outpatient appointment). These costs were inflated to the 2020/21 values using the NHS Cost Inflation Index (NHSCII) for Pay & Prices. Also, private expenses by patients reported at the time of the survey were inflated to 2020/21 prices. We used wage data from 2021 to match the date of the direct cost data. We added these details to Methods/Valuing resource use to derive costs where the years were missing.

R.1.10. For the qualitative study (page 10, lines 35-42), what was the response rate? What was the mean and median age of participants?

Mean age of participants was 49.19, median age was 50.5 Of the 19 patients that were approached 3 declined to participate. We have added these details to the paper. (See Results/Sample)

R.1.11. Discussion - page 15, line 9 - first sentence may need rewording "The found...".

Thanks for pointing out this typo, we have corrected it. "We found.."

R.1.12. Lines 16-20 - the qualitative results may be overstated as it is difficult to compare personal opinions. The sample size for the qualitative study should be indicated here again (n=...).

We have added the sample size and rephrased the sentence:

'Qualitative interviews with sixteen participants complemented the quantitative analysis to better understand patients' perspectives. Lower productivity losses and informal care need in the boot group were reflected in the interviews, and some participants in the boot group highlighted that they were more satisfied with their treatment option, felt more independent, and were able to return to their usual activities sooner.' (See Discussion/ Statement of principal findings)

R.1.13. Results in context - page 16 - please provide more information on the studies that have been conducted to date on this topic (e.g., sample sizes, locations, limitations) and how they compare to your results.

According to our knowledge, this is the first cost-effectiveness evaluation alongside a trial comparing boots with cast. Nevertheless, a few other studies have also compared resource use, return to work or to normal activities between functional treatment and cast groups, or reported EQ-5D-5l results. We have added the following to the Results in context section. (See Discussion/Results in context)

'Nevertheless, a few other studies have also compared resource use, return back to work or to normal activities between functional treatment and cast groups. For example, Egol et al (2000) reported earlier return to work in the brace group for patients who underwent internal fixation²⁹, however these patients were all instructed to avoid weight-bearing on the affected side for six weeks. Honignmann (2007) found that for patients who had malleolar fractures followed by open reduction and internal fixation, patients in the orthosis group with prescribed full weight bearing returned to work sooner, however the difference was not significant³⁰. Simansky et al (2006) and Lehtonen et al (2003) no significant differences in return to work

comparing functional treatment versus cast patients who underwent open reduction and internal fixation ^{11 31.}

In our study, no differences were observed in how soon participants returned to driving or full pre-injury work duties. The boot group indicated at the 6-week post-operative time point that their injury was having a greater impact on daily activities; however, this difference was no longer present 4 weeks later.

Similarly to our results, Kearney et al. (2021) and Haque et al. (2023) found no significant difference between brace and cast groups in terms of EQ 5D-5L scores at any time point ^{32 33.}

R.1.14. Page 16, line 13 - please cite the paper on the clinical results from the trial. Please also include more information in the Background and Discussion on the clinical results that were found in the trial.

The main results of the clinical paper are now summarised in the Background section (see Response to comment R.1.1. and R.1.2.)

The clinical paper has been drafted but has not been submitted for publication yet, but it is aimed to be submitted in 2023. We refer to the Final report submitted to the Funder instead.

R.1.15. Page 19 - Patient consent form - it states in this section that no person's data are included in the manuscript, however, the manuscript involves quotes from participants in the qualitative study with specific information provided (e.g., participant who mentions MTV Clashes in Plymouth).

We have removed those references to the place and event from the quote, to make sure the person is not identifiable. (See Table 4)

R.1.16. Table S1 - page 31 - please include year of costs.
For hospital procurement, when were the data obtained (e.g., year)?

The year of costs is included in the 2nd column for the items separately. For hospital procurement, we added the year (2022) when the data was obtained.

Reviewer: 2

Dr. Kentaro Amaha, St Luke's International University

Comments to the Author:

It is a well written manuscript with originality and comprehensible review of the literature. We believe that this is an interesting topic for foot and ankle surgeons and that this knowledge is useful for clinical practice. It could be beneficial for the scientific community as it is a prospective study with clear message.

We thank the reviewer for the positive feedback on the paper.

More specific comments are the following:

TITLE

No comment.

ABSTRACT

Clear and comprehensive.

INTRODUCTION

In the introduction section, there has been done a meticulous work to provide a short but adequate background of the study. Besides, the importance of the study is also designated through this section and the purpose is also clearly cited.

METHODS

The research methodology is detailed and well written.

RESULTS

The outcomes of the study are presented comprehensively. However, the results section is long, please make it more concise.

We have transferred the quotes of the qualitative interviews to a separate paper, that substantially shortens the Results section, see Table 4.

DISCUSSION

In the DISCUSSION, the content is concise and well written. Additionally, the clinical relevance of the study is well presented and the limitations of the study are plainly analyzed.

REFERENCES

No comment.

FIGURES

No comment.

TABLES

No comment.

While Cost-effectiveness analysis is a great methodology, the results (estimates) obtained from it are quite unstable and can easily change depending on assumptions. I am not that familiar with statistics so we should check with a statistician to see if there are any problems.

We agree with the reviewer that the confidence intervals of the NMB values are large when the analysis is done from the societal perspective. Nevertheless, sensitivity analysis to different assumptions confirms the robustness of the results. The analysis was done by health economists (PB, ECB and EM). The statisticians Pete Thomas and SD prepared the database for the economic analysis, provided comments on the economic analysis results, and co-wrote the joint statistical and health economics plan.

VERSION 2 – REVIEW

REVIEWER	Iwona Bielska McMaster University, Department of Health Research Methods, Evidence, and Impact
REVIEW RETURNED	12-Nov-2023

GENERAL COMMENTS	Thank you for the opportunity to review the revised version of the manuscript. I would like to acknowledge that the authors carefully addressed all of my questions and comments from the first review and provided detailed answers and additions to the manuscript, that greatly strengthened it. I have no further major comments. Minor comments: - Page 15, line 34: the word "found" seems to be missing in the Simansky sentence.- For the "Patient consent form" section on page 18, the authors write "No individual person's data are included in the manuscript." However, quotes from the qualitative interview are included in the Appendix. As such, I am not sure if this sentence should be revised or left as is. Thank you once again for giving me the chance to review your paper.
--